# Effect of Zero Growth of Fertilizer Action on Ecological Efficiency of Grain Production in China under the Background of Carbon Emission Reduction

**Zhongfang Zhang** [1,*], **Lijun Hou** [2], **Yuhao Qian** [1] **and Xing Wan** [2]

1   College of Food and Materials, Nanjing University of Finance and Economics, Nanjing 210003, China
2   School of Business Administration, Nanjing University of Finance and Economics, Nanjing 210003, China
*   Correspondence: 1920200015@stu.nufe.edu.cn

**Abstract:** At present, the problem of non-point source pollution and carbon emissions caused by excessive application of fertilizer is increasingly serious and has caused damage to the ecological environment. The "zero growth of fertilizer use by 2020 action plan" was introduced to solve the related ecological and environmental problems. Based on the panel data of 31 provinces in China from 1998 to 2020, this paper used the super efficiency SBM model to measure the ecological efficiency of grain production in China, and further verified the mediating effect of fertilizer application amount on the effect of zero growth of fertilizer on the ecological efficiency of grain production using the mediating effect model. The results showed that (1) zero growth of fertilizer action had a significant effect on the ecological efficiency of grain production. That is, the implementation of zero growth of fertilizer action can help improve the ecological efficiency of grain production. (2) The application amount of fertilizer played a mediating role in the mechanism of the effect of zero growth of fertilizer action on the ecological efficiency of grain production. (3) The implementation of the zero growth of fertilizer action effectively reduced the amount of fertilizer application and reduced fertilizer non-point source pollution and carbon emissions, which improved the ecological efficiency of grain production. According to the results of empirical research, to promote the sustainable development of agricultural production, more relevant ecological and environmental protection policies should be introduced and relevant subsidies should be increased.

**Keywords:** fertilizer non-point source pollution; carbon emissions; super efficiency SBM; fertilizer zero growth action; ecological efficiency of grain production

## 1. Introduction

Since the 18th National Congress of the Communist Party of China, grain yield has grown steadily in China and ranks first in the world under the guidance of the national grain security strategy and the new grain security concept. From the perspective of grain production under the new situation, insufficient output is no longer the main contradiction of agriculture, and increasingly prominent ecological problems have become the bottleneck restricting the sustainability of China's grain security. According to statistics, in 2013, China's fertilizer input exceeded 59.12 million tons, and the fertilizer application intensity reached 328.5 kg/ha, far higher than the world average (120 kg/ha); 2.5 times that of the European Union and 2.6 times that of the United States [1]. The chemical oxygen demand (COD), total nitrogen (TN) and total phosphorus (TP) of the three major pollutants in China's agricultural production reached 13.24 million tons, 2.7 million tons and 284,700 tons, respectively [2]. Agricultural non-point source pollution has surpassed industrial pollution to become the largest source of pollution. According to a report in 2014, soil pollution exceeded 16.1%, 82.8% of which was inorganic pollution [3]. In view of the urgent situation of agricultural ecology, to take the modern agricultural development road of resource saving and environmental friendliness, in 2015 the Ministry of Agriculture formulated

the "Zero growth of fertilizer use by 2020 action plan" [1], trying to solve the problem of excessive and unreasonable use of fertilizer in agricultural production. According to the data of "China Rural Statistical Yearbook" and using relevant research methods [4] to calculate, the amount of fertilizer applied in China's grain production has decreased from 40.405 million tons in 2015 to 35.852 million tons in 2020, a decrease of 11.2%, as shown in Figure 1a. Non-point source pollution of fertilizers and carbon emissions of fertilizers in the process of grain production also showed a gradual decline, as shown in Figure 1b,c.

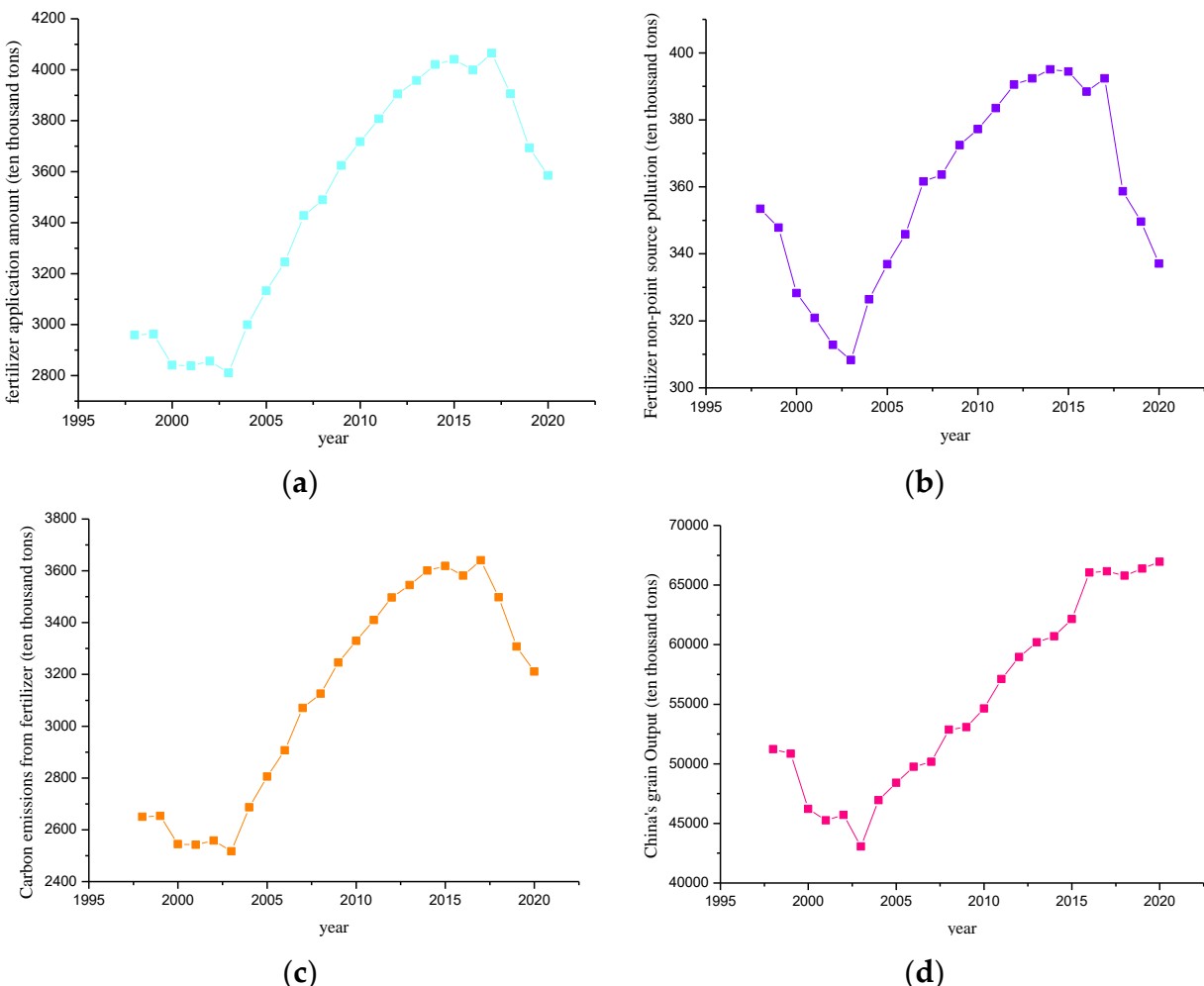

**Figure 1.** (**a**) Temporal trend of fertilizer application in grain production; (**b**) Temporal trend of fertilizer non-point source pollution application in grain production; (**c**) Temporal trend of fertilizer carbon emissions from grain production; (**d**) Temporal trend of grain production in China.

However, grain output in China gradually increased and maintained a high level, increasing by 7.7% from 621.438 million tons in 2015 to 669.491 million tons in 2020, as shown in Figure 1d. While people are concerned about grain production and want to reduce agricultural pollution, these two issues are interrelated and contradictory, and conclusions drawn from either side may be one-sided. Only realizing the coordinated development of the two can fundamentally solve this problem. The ecological efficiency of grain production is a comprehensive indicator of grain production and the ecological environment. It pays attention to both grain output and ecological problems. Therefore, in this context, the question is whether the ecological efficiency of grain production has improved? Research on this issue can not only test the achievements of the zero growth of fertilizer in the sustainable development of agriculture, but also provide a reference for national policy in the field of ecological protection.

The literature has deeply studied the ecological efficiency of grain production and its influencing factors. First, the measurement of ecological efficiency mainly includes the differences in measuring methods and research contents. The measurement methods of ecological efficiency mainly include ecological footprint analysis, stochastic frontier analysis, the DEA model and the SBM model. For example, Fu et al. (2013) used the ecological footprint index method to predict grain security and ecological sustainable development in China in 2030 [4]. Zhou et al. (2022) studied the impact of rice planting system on the environment from the perspective of ecological footprint [5]. Tian et al. (2016) used stochastic frontier analysis to measure the environmental efficiency of farmers' production in China [6], and Zhang et al. (2019) used stochastic frontier analysis to calculate the green total factor productivity (GTFP) of countries along the Belt and Road [7]. You et al. (2016), Pang et al. (2016) and Lu et al. (2021) used the DEA model to measure agricultural ecological efficiency [8–10]. Xing et al. (2019), Li et al. (2022), Tian et al. (2019) and Yao et al. (2021), Wu et al. (2022) used the SBM model and super-efficiency SBM model, respectively, to measure ecological efficiency [11–15]. Ecological footprint analysis was simple and applicable, but there were uncertainties caused by data quality, key parameters, and inherent defects of the model [16]. Stochastic frontier analysis could only deal with a single output problem, but it was difficult to take ecological impacts into account so it was rarely used to calculate ecological efficiency [17]. The traditional radial DEA model could not consider the influence of slack variables on the efficiency value, so the evaluation results may be inaccurate or biased [18]. The non-radial and non-angular SBM model proposed by Tone [19] added slack variables to the objective function, which solved the problem of slack in input and output in efficiency evaluation when there was undesired output, and provided a more accurate efficiency measure. On this basis, the super-efficiency SBM model could solve the situation that the efficiency value of multiple DMU was 1 in the calculation of the traditional SBM model. The research content is mainly regarding the national, regional, or provincial levels. Chen et al. (2019), Lu et al. (2020), Xia et al. (2021) and Su et al. (2022) [20–23] studied ecological efficiency at the national level. Based on the super-efficiency SBM model and Tobit model, Li et al. (2020) compared the green production efficiency of the three functional grain areas and found that there were significant differences among them. The green production efficiency of grain in the main grain producing areas and the production and sales balance areas were much higher than those in the main grain marketing areas [24]. Tu et al. (2015) assessed the ecological environment of rice in the Mekong Delta region of Vietnam [25]. Hu et al. (2022) analyzed the spatiotemporal heterogeneity of agricultural eco-efficiency in the Yangtze River Basin in China [26]. Zhong et al. (2020) Studied the eco-efficiency of maize production in arid Northwest China [27]. Second, regarding the influencing factors of grain production ecological efficiency, research is mainly conducted at the micro and macro levels. The micro level mainly included the agricultural acreage, the scale of farmers [28], the characteristics of farmers, agricultural technology training [6], farmers' education years [29] and so on. At the macro level, it mainly included financial development [30], technological progress [31], GDP [32], agricultural informatization [33], farmland transfer [34], level of agricultural mechanization [35], urbanization level [36,37] and agricultural policy included smart agriculture, environmental regulation and low-carbon trade pilot [38–40]. In terms of the zero growth of fertilizer action, existing studies included the driving factors of fertilizer reduction [39–44] and the contribution of fertilizer to agricultural pollution reduction [45–47].

The literature on the ecological efficiency of grain production and its influencing factors is usually about the calculation of ecological efficiency and direct influencing factors, without in-depth analysis from the transmission path, and does not consider the impact of zero growth of fertilizer on the ecological efficiency of grain production in China. The studies on the zero growth of fertilizer only focus on the effects of fertilizer application and pollution reduction, and do not study the mechanism of the progressive effects of the implementation of the zero growth of fertilizer, the fertilizer application amount, or the ecological efficiency of grain production.

The marginal contribution of this paper is as follows: First, this paper takes the action of zero growth of fertilizer as the research object and treats it as a dummy variable to study the impact of the action of zero growth of fertilizer on the ecological efficiency of grain production, which expands the research horizon of the influencing factors of ecological efficiency and provides a new path for the improvement of ecological efficiency. Second, this study uses the mediating effect model to explore the mechanism of the impact of zero growth of chemical fertilizer on the ecological efficiency of grain production, and empirically tests the transmission path of the two which can deepen the understanding of the relationship between the two. Third, this study examines the impact of action of zero growth fertilizer on the ecological efficiency of grain production in different regions to investigate the heterogeneity of the impact of zero growth fertilizer action and analyze the reasons for different results to provide different improvement measures for different regions.

## 2. Materials and Methods

### 2.1. Data Sources

Considering the data integrity and availability, this article selects 1998–2020 panel data of 31 provinces in China as the research object. All the data from the "China Statistical Yearbook", "China Rural Statistical Yearbook", "China Financial Yearbook", "China Population and Employment Statistics Yearbook" and Statistical Yearbook of each province, and the interpolation method is used to supplement the partially missing data.

### 2.2. Definition of Variables

#### 2.2.1. Dependent Variables

In this paper, the ecological efficiency of grain production (lnEco) is taken as the explained variable and calculated by the super-efficiency SBM model. It reflects the degree of resource utilization and environmental protection in the process of grain production.

Suppose that the system contains $n$ decision units, $N$ input indicators, $M$ expected output indicators and $L$ unexpected output indicators, which are represented by vectors $x \in S^N$, $y^a \in S^M$, $y^b \in S^L$, where $x$, $y^a$ and $y^b$ are all matrices $x = [x_1 \ldots x_n] \in S^{N \times n}$, $y^a = [y_1^a \ldots y_n^a] \in S^{M \times n}$, $y^b = [y_1^b \ldots y_n^b] \in S^{L \times n}$.

The model construction of the super-efficiency SBM is as follows:

$$\rho^* = \min \frac{1 - \frac{1}{N} \sum\limits_{n=1}^{N} \frac{s_n^x}{x_{kn}^t}}{1 + \frac{1}{M+I} \left( \sum\limits_{m=1}^{M} \frac{s_m^y}{y_{km}^t} + \sum\limits_{i=1}^{L} \frac{s_i^b}{b_{ki}^t} \right)} \quad (1)$$

$$s.t. \begin{cases} \sum\limits_{k=1,k \neq j}^{N} z_k^t x_{kn}^t + s_n^x = x_{kn}^t, n = 1, \cdots, N \\ \sum\limits_{k=1,k \neq j}^{N} z_k^t y_{km}^t + s_m^y = y_{km}^t, m = 1, \cdots, M \\ \sum\limits_{k=1,k \neq j}^{N} z_k^t b_{ki}^t + s_i^b = b_{ki}^t, i = 1, \cdots, I \\ z_k^t \geq 0, s_n^x \geq 0, s_m^y \geq 0, s_i^b \geq 0, k = 1, \cdots, K \end{cases} \quad (2)$$

where $x_{kn}^t$, $y_{km}^t$ and $b_{ki}^t$ respectively represent the actual values of input factors, expected output and unexpected output of the period t producer decision unit ***K***. $S_n^x$, $S_m^y$ and $S_i^b$ represent the relaxation variables of the input variables, expected outputs and unexpected outputs, respectively. $z_k^t$ represents the weight of the decision-making unit.

This study designed indicators to measure the ecological efficiency of grain production from three aspects: input, expected output and unexpected output. In terms of the input variables, labor force, land, fertilizer, pesticide, agricultural film, effective irrigated area and total power of machinery were selected as indicators. In terms of the output variables,

grain yield was selected as the expected output indicator, and total carbon emissions and fertilizer non-point source pollution in grain production were selected as two unexpected output indicators (Table 1).

**Table 1.** Selected input–output indicators.

| Index | Variable | Variable Description (Unit) |
|---|---|---|
| Input variables | Land input | Grain sown area (thousands of hectares) |
| | Labor input | Number of employees in first industry $\times \alpha \times \beta$ (ten thousand people) |
| | Fertilizer input | Total input of fertilizer $\times \beta$ (ten thousand tons) |
| | Pesticide input | Total input of pesticide $\times \beta$ (ten thousand tons) |
| | Agricultural film input | Total input of agricultural film $\times \beta$ (ten thousand tons) |
| | Effective irrigation area | Total effective irrigated area $\times \beta$ (thousands of hectares) |
| | Total mechanical power | Total power of agricultural machinery $\times \beta$ (ten thousand kilowatts) |
| Expected output | Grain output | Grain yield (ten thousand tons) |
| Unexpected output | Carbon emission | Carbon emissions from fertilizer, pesticide, agricultural film, diesel oil, plowing and irrigation are added together (ten thousand tons) |
| | Fertilizer non-point source pollution | Total nitrogen loss + Total phosphorus loss (ten thousand tons) |

Note: $\alpha$ = agricultural output value/total output value of agriculture, Forestry, Animal husbandry and Fishery. $\beta$ = grain sown area/total sown area of crops.

Among the input indicators, the data of grain sown area can be obtained directly, while other indicators cannot. Other indicators can be obtained from the total input of agricultural production by referring to other literature [4]. The undesired output index is divided into carbon emissions in grain production and non-point source pollution of fertilizer. The carbon emission coefficient in grain production refers to the index coefficient of Li et al. (2011) [48]: 0.8956 kg/kg for fertilizer, 5.18 kg/kg for agricultural film, 4.9341 kg/kg for pesticide and 0.592 kg/kg for diesel, irrigation was 20.476 kg/hm$^2$, and agricultural cultivation was 312.6 kg/hm$^2$. In terms of indicators of fertilizer non-point source pollution, the main non-point source pollution in grain production comes from fertilizer. The calculation method refers to the unit list method of Lai et al. [49], and the relevant loss coefficient refers to Lu et al. (2020) [19], who adopted the "Manual of Fertilizer Loss Coefficient of Agricultural Pollution Sources in the First National Pollution Source Census".

### 2.2.2. Explanatory Variable

In this paper, the zero growth of fertilizer action (Zer) was taken as the explanatory variable to explore the impact of this action on the ecological efficiency of grain production. Since the zero growth of fertilizer action was launched in 2015, 2015 was taken as the cutoff point in this study, and the previous year was marked as 0 and the subsequent year as 1.

### 2.2.3. Mediating Variables

The fertilizer application amount (lnFer) in grain production is taken as a mediating variable.

### 2.2.4. Control Variables

The ecological efficiency of grain production is also affected by the urbanization rate (lnUrb), financial expenditure on agriculture (lnFin), the proportion of the first industry (lnPro), farmers' income (lnFar), per capita sown area of crops (lnPer), etc.

The urbanization rate reflects the degree of urbanization. In the process of urbanization, farmland is occupied, and the rural labor force begins to shift, which competes with agricultural production factors and affects grain production efficiency. Because the statistical caliber changed, the variables of financial expenditure on agriculture also changed. From 1998 to 2002, it was "expenditure on supporting agricultural production+ expenditure on comprehensive agricultural development+ expenditure on agriculture, forestry, water

conservancy and meteorology". From 2003 to 2006, it was "expenditure on agriculture + expenditure on forestry + expenditure on water conservancy and meteorology". After 2007, it was expressed as "expenditure on agriculture, forestry and water conservancy". On the one hand, the increase in financial expenditure on agriculture will enable farmers to buy more inputs such as fertilizers and pesticides. On the other hand, it will increase infrastructure construction and increase the scale of grain cultivation. The proportion of the first industry refers to the proportion of the output value of the first industry in the total output value, reflecting the degree of economic development of a region. If the proportion of the first industry is higher, it pays more attention to grain production and has a relatively rich labor force, but it may not have a strong awareness of ecological environmental protection. The farmers' income is expressed as the per capita disposable income of rural residents, with the farmers' disposable income levels increasing. On the one hand, the ability of farmers to buy agricultural machinery, fertilizers, and pesticides improves, and they can also buy compound fertilizer whose pollution is smaller while price is higher. On the other hand, with the improvement of income level, people's environmental protection consciousness will also strengthen gradually. The per capita sown area of crops is expressed by the ratio of the sown area of grain to the rural population. The sown area of crops per capita reflects the scale of grain production, which can save costs and improve production efficiency in mechanized operations.

### 2.3. Mediating Effects Model

Based on existing studies, this paper uses the stepwise test regression coefficient method proposed by Wen et al. [50] for reference. To eliminate the influence of heteroscedasticity, all variables except the variable "whether to implement zero growth of chemical fertilizer action" are taken logarithms to construct the mediation effect model and test:

$$\ln Y_{it} = \alpha_0 + \alpha_1 action_{it} + \sum \alpha_j \ln z_{ijt} + \varepsilon_{1it} \tag{3}$$

$$\ln M_{it} = \beta_0 + \beta_1 action_{it} + \sum \beta_j \ln z_{ijt} + \varepsilon_{2it} \tag{4}$$

$$\ln Y_{it} = \gamma_0 + \gamma_1 action_{it} + \gamma_2 \ln M_{it} + \sum \gamma_j \ln z_{ijt} + \varepsilon_{3it} \tag{5}$$

where $i$ and $t$ represent the province and year, respectively, and $Y_{it}$ is the dependent variable, representing the ecological efficiency value of grain production in the t year of province $i$. $M_{it}$ is the mediating variable, which is the amount of fertilizer applied in grain production, and action is the key explanatory variable indicating whether to implement the zero growth of fertilizer action. $z_{ijt}$ represents control variables, including urbanization rate, financial expenditure on agriculture, proportion of the first industry, farmers' income and per capita sown area of crops. $\alpha_0, \alpha_1, \alpha_j, \beta_0, \beta_1, \beta_j, \gamma_0, \gamma_1, \gamma_2, \gamma_j$ are the parameters to be estimated. $\varepsilon_{it}$ represents the random error term. A stepwise regression test was used to test the mediating effect. $\alpha_1$ is the total effect of zero growth of fertilizer action on the ecological efficiency of grain production. $\beta_1$ is the effect of zero growth of fertilizer action on the amount of fertilizer applied in grain production. $\gamma_1$ is the direct effect of zero growth of fertilizer action on ecological efficiency after controlling the effect of the amount of fertilizer applied. $\gamma_2$ is the effect of the mediating variable on the ecological efficiency of grain production after controlling for the effect of zero growth of the fertilizer action. The mediating effect is equal to $\beta_1 \times \gamma_2$, and the proportion of the mediating effect is $\beta_1 \times \gamma_2 / \alpha_1$. The Sobel test and bootstrap test were used to ensure the accuracy of the test results.

## 3. Results

### 3.1. Measurement of the Ecological Efficiency of Grain Production

Based on MaxDEA 8.0 software and using the super-efficiency SBM model, this study calculates the ecological efficiency of grain production in 31 provinces from 1998 to 2020.

The ecological efficiency in this study refers to the comprehensive efficiency. As shown in Figure 2, the ecological efficiency of grain production fluctuated during 1998–2020,

generally showing a trend of first declining, then stabilizing and rising. From 1998 to 2009, it essentially showed a downward trend, and from 2009 to 2015, it showed a relatively stable state. From 2015, it showed a rapid upward trend, indicating that the utilization and allocation efficiency of resources in grain production significantly improved after 2015. The change in trend of pure technical efficiency was consistent with that of ecological efficiency. The difference was that the pure technical efficiency showed an upward trend from 2001 to 2004, and the trend was consistent in other periods. Especially after the implementation of the zero growth of fertilizer action in 2015, the pure technical efficiency also rose sharply. The application of technology in most provinces of grain production in China is gradually rationalized. In terms of scale efficiency, the overall scale efficiency of grain production in China was at a relatively high level from 1998 to 2020, with the lowest value remaining at approximately 0.9, indicating that the input proportion of various factors in grain production in China was relatively appropriate, and the sown area and other factors remained at a relatively fixed level.

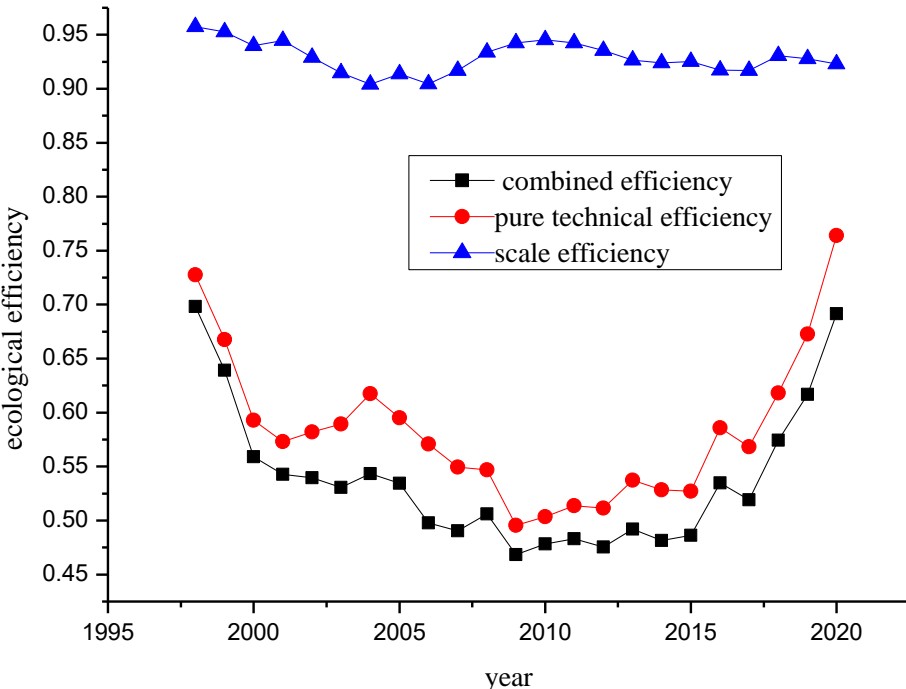

**Figure 2.** Provincial mean trend of ecological efficiency, pure technical efficiency and scale efficiency.

Figure 3a shows the temporal trend of the mean ecological efficiency in the three grain functional areas. Longitudinally, the mean value of ecological efficiency of major grain-producing areas and grain production and sales balance areas are higher than those of main grain sales areas. Before 2010, the mean value of ecological efficiency of grain production and sales balance areas was higher than that of major grain producing areas. After 2010, the mean value of ecological efficiency of major grain producing areas was higher than that of grain production and sales balance areas. Horizontally, the mean value of ecological efficiency in major grain-producing areas fluctuated, first decreasing and then increasing, and the value increased from 0.696 in 1998 to 0.747 in 2020. It can be seen that provinces in major grain producing areas attached more importance to the ecological environment, especially in the "National Main Function Plan" issued by The State Council in 2010. It was proposed that the main grain producing areas were also the most important ecological barrier areas in China, and they shouldered the main function of grain security and ecological security in China. The main grain-producing area gradually played a key role in the ecological barrier function after 2010. In general, the mean value of ecological efficiency in the grain production and sales balance areas showed a trend of first declining

and then increasing, especially from 2017, when it increased from 0.517 to 0.762 in 2020, with an annual growth rate of 8.17%. It can be seen that the ecological environment protection in the grain production and sales balance areas has gradually begun to be considered, and significant achievements have been reached. The mean value of ecological efficiency in main grain sales areas was much lower than that of the major grain producing areas and the production and marketing balance areas, which may be due to the different main body functions. The main grain sales areas were not as good in the other two areas in terms of geographical location, climate conditions, degree of agricultural technology, and state support for agriculture, which led to a low overall grain production efficiency.

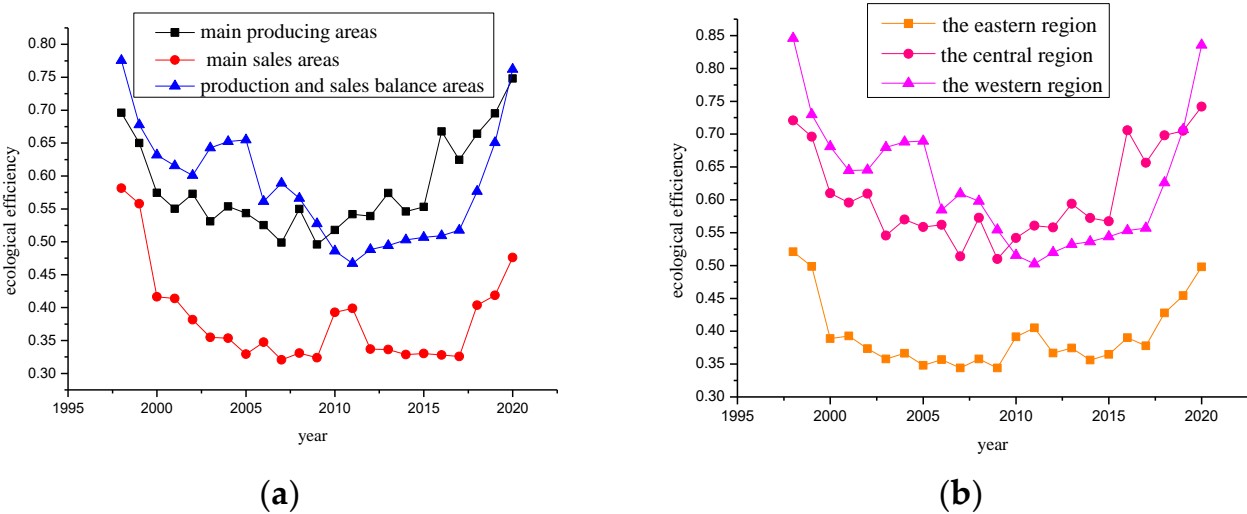

**Figure 3.** (**a**) Mean time trend of ecological efficiency in three grain functional areas; (**b**) Average time trend of ecological efficiency in the three regions in China.

As shown in Figure 3b, the mean values of grain production ecological efficiency in the eastern, central, and western regions all showed a trend of first declining, then steadying and finally rising, especially after 2015. The mean value of ecological efficiency of the western and eastern regions was higher than that of the eastern regions, and before 2010, the mean value of ecological efficiency of the western regions was higher than that of the central regions. The mean value of ecological efficiency of the central regions was higher than that of the western regions after 2010; until 2019, the western regions surpassed the central regions again. The mean value of ecological efficiency in the eastern region was significantly lower than that in the other two regions, possibly because its economy was relatively developed and more attention was given to rapid development, resulting in neglected agriculture. At the same time, the second and third industries were developed, and the labor force was mainly concentrated on them, so the local labor opportunity cost was larger. Farmers also overused chemical fertilizers, pesticides, and other elements to replace labor, which resulted in serious environmental problems in the process of grain production.

*3.2. The Impact of Zero Growth of Fertilizer Action on the Ecological Efficiency of Grain Production*
3.2.1. Banmark Regression

Using Stata 15.0 software, the Fisher-ADF unit root test was first performed on variables to prevent the phenomenon of false regression, and the null hypothesis that the panel contained unit roots was strongly rejected by all statistics throughout the test. Second, the Hausman test was conducted to determine whether to choose the fixed effect model or the random effect model. The original hypothesis was rejected by the *p* value test, so the fixed effect model was selected.

It can be seen from the regression results in Table 2 that the implementation of zero growth of fertilizer action has a significant positive effect on the ecological efficiency of grain production at the 1% significant level; that is, the zero growth of fertilizer action improves the ecological efficiency of grain production. The action reduced the phenomenon of excessive and unreasonable use of fertilizer in the process of grain production, thus reducing the non-point source pollution load and the corresponding fertilizer carbon emissions [14]. To a certain extent, the action reduced the expected output but did not reduce the expected output, improving the ecological efficiency of grain production on the whole.

**Table 2.** Empirical estimation results of the impact of zero growth of fertilizer action on the ecological efficiency of grain production.

| Types of Variables | Variables | Coefficient | Standard Error |
|---|---|---|---|
| explanatory variable | Zer | 0.0909 *** | 0.0285 |
| | lnUrb | −0.3451 *** | 0.0484 |
| | lnFin | −0.1548 *** | 0.0277 |
| | lnPro | −0.0407 | 0.0348 |
| control variable | lnFar | 0.2610 *** | 0.0588 |
| | lnPer | 0.3528 *** | 0.0440 |
| | constant term | −0.2246 | 0.2512 |
| R-squared = 0.2663, F(6667) = 40.88, Prob > F = 0.0000 | | | |

Note: *** indicates significance at the levels of 1%.

The urbanization rate and financial expenditure on agriculture both have a significant negative effect on the ecological efficiency of grain production at the 1% significance level because with the improvement of the urbanization rate, it will form a competitive situation for grain production environment elements, such as decreasing the area of arable land and the number of laborers. The increase in financial expenditure on agriculture enables people to buy more inputs, such as fertilizers, pesticides, and agricultural films, and excessive input of these factors will cause the destruction of arable land, non-point source pollution and carbon emissions to a certain extent. Farmers' income and per capita sown area have a significant positive effect on the ecological efficiency of grain production both at the 1% significance level; that is, with the increase in farmers' income, they can buy more agricultural machinery and tools and improve their own technical production knowledge, which improves the scale of production and at the same time, their awareness of environmental protection is also enhanced. The increase in per capita sown area is also a sign of agricultural scale, which is conducive to operating mechanistically, saving costs, and improving production efficiency. The influence of the proportion of first industry on the ecological efficiency of grain production is negative but not significant.

### 3.2.2. Heterogeneity Analysis: Subregional Regression

To consider the impact of zero growth of fertilizer action on the ecological efficiency of grain production in different regions, this paper conducted subsample regression in the eastern, central, and western regions. Table 3 shows the regression results. We can see that zero growth of fertilizer action in different regions has a distinct influence on the ecological efficiency of grain production. There is a significant positive correlation between the two in the central and western regions. In eastern China, the effect of zero growth of fertilizer action on the ecological efficiency of grain production is positive, but the relationship between them is not significant. This is because in eastern China, the economy is more developed, so the emphasis on agriculture is less than that of the secondary and tertiary industries. The farmers here do not have strong awareness of environmental protection in agricultural production, which results in the effect of zero growth of fertilizer action on the ecological efficiency of grain production not being significant.

**Table 3.** Regression results of zero growth of fertilizer action on ecological efficiency of grain production in eastern, central and western China.

| Variables | Eastern China | | Central China | | Western China | |
|---|---|---|---|---|---|---|
| | Coefficient | Standard Error | Coefficient | Standard Error | Coefficient | Standard Error |
| Zer | 0.0778 | 0.0530 | 0.1011 ** | 0.0398 | 0.0923 ** | 0.0448 |
| lnUrb | −0.5898 *** | 0.0840 | −0.2493 *** | 0.0915 | −0.1248 * | 0.0743 |
| lnFin | −0.0227 | 0.0566 | −0.1552 *** | 0.0403 | −0.1964 *** | 0.0396 |
| lnPro | −0.0639 | 0.0609 | −0.0362 | 0.0529 | 0.1833 ** | 0.0801 |
| lnFar | 0.0909 | 0.1220 | 0.3334 *** | 0.0835 | 0.3034 *** | 0.0829 |
| lnPer | 0.5875 *** | 0.0760 | 0.0577 | 0.0622 | 0.3286 *** | 0.0832 |
| constant | −0.3969 | 0.5006 | −0.3393 | 0.4048 | −1.1325 | 0.5148 |
| $R^2$ | 0.3317 | | 0.3548 | | 0.3569 | |
| Prob > F | Prob > F(6236) = 0.0000 | | Prob > F(6170) = 0.0000 | | Prob > F(6258) = 0.0000 | |
| obs | 253 | | 184 | | 276 | |

Note: *, ** and *** indicate significance at the levels of 10%, 5% and 1%, respectively.

### 3.3. Mediating Effect Analysis

Table 4 describes the mechanism of the impact of zero growth of fertilizer action on the ecological efficiency of grain production when the amount of fertilizer applied in grain production is taken as the mediating variable. Model (2) tests the effect of zero growth of fertilizer action on the fertilizer application amount in grain production. Table 4 shows that the influence of the zero growth of fertilizer action on the fertilizer application amount in grain production is significantly negative at the 1% significance level, because the implementation of the action affects the amount of fertilizer used in grain production. Meanwhile, all other control variables have significant effects except for the farmers' income. In Model (3), all variables, including mediating variables, are entered into the regression equation to test the impact of zero growth of chemical fertilizer action on the ecological efficiency of grain production through the mediating effect of fertilizer application amount on grain production. As seen from the results in Table 4, the influence coefficient of fertilizer application amount as a mediating variable on the ecological efficiency of grain production is −0.2302, which passes the significance level test of 1%. The influence of zero growth of fertilizer action on the ecological efficiency of grain production is 0.0596, which is smaller than the coefficient 0.0909 when no mediating variable is added. At the same time, the significance level changed from 1% to 5%, so the amount of fertilizer application played a part in the mediating effect. In general, the total mediating effect of fertilizer application amount on the ecological efficiency of grain production was (−0.1359) × (−0.2302) = 0.03128, accounting for 34.42%.

**Table 4.** Zero growth action plan of fertilizer, fertilizer application amount and ecological efficiency of grain production.

| Variables | Model (2) Regression | | Model (3) Regression | |
|---|---|---|---|---|
| | Coefficient | Standard Error | Coefficient | Standard Error |
| Zer | −0.1359 *** | 0.0260 | 0.0596 ** | 0.0284 |
| lnFer | | | −0.2302 *** | 0.0411 |
| lnUrb | −0.1027 ** | 0.0442 | −0.3688 *** | 0.0475 |
| lnFin | 0.1539 *** | 0.0253 | −0.1194 *** | 0.0279 |
| lnPro | 0.4300 *** | 0.0318 | 0.0582 | 0.0384 |
| lnFar | 0.0818 | 0.0538 | 0.2798 *** | 0.0577 |
| lnPer | 0.1232 *** | 0.0402 | 0.3811 *** | 0.0433 |
| constant | 0.4671 ** | 0.2295 | −0.1170 | 0.2465 |
| N | 713 | | 713 | |
| $R^2$ | 0.4228 | | 0.2987 | |
| Prob > F | Prob > F(6676) = 0.0000 | | Prob > F(7675) = 0.0000 | |

Note: ** and *** indicate significance at the levels of 5% and 1%, respectively.

The Sobel test and bootstrap test were performed to determine the mediating effect of the fertilizer application amount more accurately. The Sobel test results showed that the *p* value was 0.013, and the null hypothesis was rejected at the 5% level, indicating that the mediating effect was significant and the proportion of the mediating effect was 15.2%. The bootstrap test results are shown in Table 5. It can be seen that both direct and indirect effects are significant. Meanwhile, in the 95% confidence interval, whether before or after deviation correction, the confidence interval of both direct and indirect effects does not include 0 so the mediating effect is significant.

**Table 5.** Mediating effect results tested by the bootstrap method.

| Mediating Variable | Influencing Mechanism | Effect Estimates | Standard Error | 95% Confidence Interval | | 95% Confidence Interval after Deviation Correction | |
|---|---|---|---|---|---|---|---|
| | | | | Lower Limit | Upper Limit | Lower Limit | Upper Limit |
| lnFer | indirect effect | 0.0313 ** | 0.0130 | 0.0058 | 0.0568 | 0.0080 | 0.0596 |
| | direct effect | 0.1753 *** | 0.0399 | 0.0970 | 0.2536 | 0.0971 | 0.2556 |

Note: ** and *** indicate significance at the levels of 5% and 1%, respectively.

### *3.4. Robustness Test*

To analyze the robustness of the above results, this paper continues to use a random effects model (RE) and the instrumental variable method (IV) with the first-order lag term of independent variables as instrumental variables to perform regression on the mediating effects model. Table 6 shows that the results obtained by different estimation methods are basically the same, which indicates that the research conclusions of this paper are robust.

**Table 6.** Robustness test of the ecological efficiency of grain production.

| Variable | First Stage | | Second Stage | | Third Stage | |
|---|---|---|---|---|---|---|
| | RE | IV | RE | IV | RE | IV |
| Zer | 0.1123 *** (0.0278) | 0.1260 *** (0.0262) | −0.1248 *** (0.0268) | −0.1598 *** (0.0243) | 0.0988 *** (0.0275) | 0.0937 ** (0.0266) |
| Control variables | yes | yes | yes | yes | yes | yes |
| lnFer | | | | | −0.1492 ** (0.0327) | −0.2019 *** (0.0416) |
| Control variables | | | | | yes | yes |

Note: ** and *** indicate significance at the levels of 5% and 1%, respectively.

## 4. Discussion

This study analyzed the impact of zero growth of fertilizer on the ecological efficiency of grain production in China, and tested the mediating effect of fertilizer applied amount on this impact. The significance of this study lies in how to better deal with the issue of grain production and environmental protection under the background of emphasizing carbon emission reduction and ecological environmental protection. The research content and results analysis of this paper mainly focus on the following aspects.

### *4.1. Measurement of Ecological Efficiency on Grain Production*

Ecological efficiency of grain production was a comprehensive index of input, expected output, and unexpected output. Since the implementation of the zero growth of fertilizer action in 2015, the amount of fertilizer used in grain production had decreased, and non-point source pollution and carbon emissions generated by fertilizer had also decreased, as shown in Figure 1. Since the super-efficiency SBM model could fully solve the problem of slack variables in the DEA model when calculating the efficiency with unexpected output, therefore, we adopted the super-efficiency SBM model to calculate the ecological efficiency

of grain production from 1998 to 2020 in China. The Figure 2 showed that during this period, the ecological efficiency of grain production first decreased and then stabilized, and showed an obvious upward trend after 2015 which indicated that the input-output efficiency of grain production was higher after 2015. On the one hand, the utilization rate of input resources was improved but on the other hand, the grain output constantly increased and the unexpected output constantly decreased. Figure 3a,b showed the ecological efficiency results of grain production in the three grain functional areas and eastern, central, and western regions respectively. In the three grain functional areas, the ecological efficiency of grain production in the main sales areas was significantly lower than that of main producing areas and production and sales balance areas. Correspondingly, the ecological efficiency of the eastern region was obviously lower than that of the central and western regions. The provinces in the main sales areas are located in the eastern region whose economies were developed; they did not pay enough attention to the grain yield and the degree of environmental protection in the production process. At the same time, they invested a lot of others factors to save labor costs, which leading to the increase of unexpected output.

### 4.2. Impact of the Zero Growth of Fertilizer Action on Ecological Efficiency

The Table 2 shows the fixed effect model test results, where the effect of zero growth of fertilizer action on the ecological efficiency of grain production was significantly positive at the significance of 1%, with a coefficient of 0.0909. Fertilizer zero growth action was a part of environmental regulation in the process of green agricultural development. Based on the "porter hypothesis", moderate environmental regulation could encourage innovation [51], thus enhancing resource utilization. The innovation compensation effect could improve grain production efficiency and promote the ecological efficiency of grain production. Agricultural non-point source pollution was external and uneconomical, so environmental regulation became a necessary means for the government to control agricultural non-point source pollution. The control and prevention of agricultural pollution could not be achieved without an effective institutional environment and relatively perfect institutional design and implementation [52]. At the same time, the zero growth of fertilizer action could lead to induced technological change [53]. For example, it could promote the technological progress of grain production, or the continuous updating of waste recycling technology, which could drive the grain production mode from traditional production to green production, achieve a reduction in agricultural non-point source pollution, and then drive the green development of grain production.

### 4.3. Impact of Zero Growth of Fertilizer Action on the Fertilizer Applied Amount

Model (2) in Table 4 showed that the zero growth of fertilizer action had a significantly negative effect on fertilizer applied amount at the significance level of 1%, with a coefficient of 0.14. Environmental pollution in grain production directly stemmed from the behavior of its stakeholders [54]. The appropriate amount of fertilizer applied in grain production was mainly affected by the stakeholders of the government, fertilizer production enterprises, and farmers. The influencing factors included macro and micro aspects, macro aspects including relevant national policies, fertilizer prices [55], the popularization degree of agricultural technology, labor input, and the level of regional economic development [56]. Micro aspects included farmers' fertilization experience, fertilization technology [57], and cognitive level [58]. As a related environmental regulation policy issued by the government, the zero growth of chemical fertilizer action was promoted to reduce the amount and increase the efficiency of fertilizer from the macro aspect and continuously promote soil testing formula technology, change the fertilization method, and increase the application of organic fertilizer. The zero growth of fertilizer action restricted the output of chemical fertilizer manufacturers to some extent. Under the promotion of the policy, the total amount of fertilizer production will decrease, leading to the rise of chemical fertilizer prices and a reduction in the amount of fertilizer used by farmers. Farmers' behavior in fertilizer application was influenced by their cognitive level, including their cognition of

national policies and fertilization technologies. Different cognitive levels led to different fertilization intentions and behaviors of farmers, namely, their cognitive changed intentions and willingness changed behaviors [59]. The implementation of the zero growth of fertilizer action could reduce the amount of fertilizer by improving farmers' fertilization technology and their awareness of environmental protection from the micro aspect. Therefore, the zero growth of fertilizer action reduced the amount of fertilizer used by the government, fertilizer producers and farmers.

### 4.4. Testing of the Mediating Effect

The results of Model (3) in Table 4 showed that after the addition of mediating variable, the influence coefficient of zero growth of fertilizer action on the ecological efficiency of grain production decreased from 0.0909 to 0.0596, which was significant at the significance level of 5%. The mediating effect was significant, and accounting for 34.42% it passed the Sobel test and Bootstrap test. As a mediating variable, the effect of fertilizer applied amount on the ecological efficiency of grain production was significantly negative, and the coefficient was 0.2302. Reducing the fertilizer applied amount could significantly improve the ecological efficiency of grain production. The output of the ecological efficiency of grain production was composed of expected and unexpected outputs. The higher the ecological efficiency of grain production was, the higher the expected output or the lower the unexpected output. In terms of expected output, fertilizer inputs must follow the law of diminishing marginal production; that was, when the technical level and other inputs were under a certain condition, grain production had a critical point. Before this, increasing fertilizer would appropriately increase marginal production; more than this, increasing fertilizer might decrease marginal production until it became negative. From the perspective of economics, scholars believed that the application amount of chemical fertilizer in China had exceeded the optimal application amount [60]. In terms of the expected output, farmers were "rational economic man"; their behaviors were to realize the maximization of self-interest, they would only consider more fertilizers that could increase grain production, and they would not take the environment into account. On the one hand, fertilizer as a grain production input affected the expected output; on the other hand, it also affected the unexpected output as a source of carbon emissions and non-point source pollution. In the current situation, the fertilizer amount reduction would not reduce the output of grain but would reduce non-point source pollution, in which way to improve the ecological efficiency of grain production.

### 4.5. Limitations and Further Work

This study has some limitations, and further work needs to be done in these areas. First of all, the nationwide data of planting industry was relatively general, and there were no special statistics on the input data of grain crops. Therefore, in the process of research, this paper used the methods of other researchers to analyze and strip the data not available to departments from the production data of agriculture, and then sorted out for use. It is expected that there will be higher quality data in the future to further verify the conclusions of this paper and analyze the ecological efficiency of grain production more comprehensively. Secondly, the mediating effect test results in this study showed that the mediating effect was partially mediated, and there may be other influence paths that play a role in the impact of zero growth of fertilizer action on the ecological efficiency of grain production, which needs to be further analyzed in future studies.

### 5. Conclusions and Recommendations

Based on the statistical data of 31 provinces in China from 1998 to 2020, this paper used the super-efficiency SBM model to measure the ecological efficiency of grain production and further verified the mediating effect of fertilizer application amount on the effect of zero growth of fertilizer action on the ecological efficiency of grain production using the mediating effect model. The main conclusions are as follows: (1) The zero growth

of fertilizer action had a significant promoting effect on the ecological efficiency of grain production, and the influence coefficient was 0.09. (2) There was a significant positive correlation between zero growth of fertilizer action and ecological efficiency of grain production in the central and western regions, but not in the eastern region. (3) The mediating effect of the fertilizer application amount was significant, and accounted for 34.42%.

Recommendations based on the conclusions of this paper are as follows: first, the government needs to introduce similar environmental protection policies, which not only enhance people's awareness of environmental protection but also make a quantitative commitment to environmental protection to improve the ecological environment. Second, the government needs to increase the intensity of financial support for agriculture and environmental protection subsidies, especially for different regions in the eastern, central and western regions, to take corresponding incentive mechanisms to facilitate the sustainable development of agriculture. Third, in the process of grain production, more attention should be given to farmers' publicity, education and, training activities to improve farmers' awareness of environmental protection and relevant knowledge levels, while enhancing farmers' initiative and enthusiasm for environmental protection and thus reducing the degree of environmental pollution.

**Author Contributions:** Investigation, L.H.; Data curation, Z.Z.; Project administration, X.W.; Funding acquisition, Y.Q. All authors have read and agreed to the published version of the manuscript.

**Funding:** This study was supported in part by National Social Science Foundation of China (grant no. 21BGL033), Humanities and Social Science Research Foundation of The Ministry of Education (grant no. 17YJAZH031), Youth Project of China Association for Science and Technology Innovation Think Tank (project no. DXB-ZKQN-2016-22), Jiangsu Graduate Research And Innovation Program (project no. KYCX21-1444), Doctoral Project of School of Food and Material Science, Nanjing University of Finance and Economics (project no. BSZX2021-01).

**Institutional Review Board Statement:** Not applicable.

**Informed Consent Statement:** Not applicable.

**Data Availability Statement:** Not applicable.

**Conflicts of Interest:** The authors declare no conflict of interest.

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
