# Peer review of "Effect of Zero Growth of Fertilizer Action on Ecological Efficiency of Grain Production in China under the Background of Carbon Emission Reduction"

_sustainability, doi:10.3390/su142215362_

Round 1
Reviewer 1 Report
The topic selection is meaningful. However, some content of this paper need to be further optimized. And I have some considerable concerns that would be worth addressing. I would like to give some suggestions as below first and if possible, give detailed suggestions for the revised paper in future.
Several problems need to be considered:
1. The paper is not traditional writing framework and lack of discussion part.
2. The comparison with related studies is not sufficient.
3. Please discuss the limitations and future directions of the paper appropriately. For instance, what factors might influence existing conclusions?
4. It would be interesting to discuss recommendations for grain production management in the context of carbon neutrality, since carbon emissions are directly related to grain production and fertilizer use;
5. The current conclusions are all conceivable and lack of innovative perspective. The paper doesn't seem to bring enough new knowledge.
Author Response
Dear Expert:
Thank you so much for your time and the positive comment for our work, your conclusion and suggestions are quite accurate, constructive and professional.
Now, we would like to answer your comments and questions one by one in the attached pdf file.
We wish our answer can clear your confusion and meet your satisfaction, and please do not hesitate to let us know if there is still some question in the revised version.
Thank you again for your help!
Best wishes!
Zhongfang Zhang

Reviewer 2 Report
overall the citation of reference very poor, discussion not support by strong references. plz add sitation to support the data in discussion
Author Response

(The authors gave the same response as above.)

Reviewer 3 Report
The article entitled Study on the effect of zero growth of fertilizer action on ecological efficiency of grain production in China is approaching an interesting topical theme. The paper qualifies by the applied methodology and validation of results to be published in Sustainability journal.
As minor changes to be clarified by authors before publication I recommend:
- To better underline in the first part the advantages represented by the application of a SBM model for their analysis
- Are there any limits imposed by the model in the analysis ? This should be explained in the limits and possible further continuation of the study
- What is the purpose of the regional analysis and what the eastern, central and western region represents in geographical terms ?
Author Response

(The authors gave the same response as above.)

Reviewer 4 Report
My revision is done and attached.

Author Response

(The authors gave the same response as above.)

Round 2
Reviewer 1 Report
Thank you for your revision. If possible, please revise the paper further according to the comments (attachment).

Author Response
Dear Expert:
Thank you so much for your time and the positive comment for our work, your conclusion and suggestions are quite accurate, constructive and professional.
Now, we would like to answer your comments and questions one by one, and the red color is the comments, the black content is our responses, the green parts are the revised content in the paper.
The attached file is the response.

Reviewer 2 Report
in the discussion it is necessary to add references to support the statement.
turnitin 24%

Author Response

(The authors gave the same response as above.)

Round 3
Reviewer 1 Report
Thank you for your revision and I’m satisfied with the current version. Of course, I have one more suggestion, which is to add secondary headings for several points in the discussion, such as 4.1, 4.2, 4.3, etc.
Author Response
Dear Expert:
Thank you so much for your time and the positive comment for our work, your conclusion and suggestion are quite accurate, constructive and professional.
Now, we would like to answer your comment and question, and the red color is the comment, the black content is our responses, the green parts are the revised content in the paper, which is the attached file.
We wish our answer can clear your confusion and meet your satisfaction, and please do not hesitate to let us know if there is still some question in the revised version.
